# Multi-Epoch Optical Spectroscopy Variability of the Changing-Look AGN Mrk 883

Erika Benítez [1,*], Castalia Alenka Negrete [1], Héctor Ibarra-Medel [1], Irene Cruz-González [1] and José Miguel Rodríguez-Espinosa [2,3]

1 Instituto de Astronomía, Universidad Nacional Autónoma de México, AP 70-264, Mexico City 04510, Mexico; alenka@astro.unam.mx (C.A.N.); hibarram@astro.unam.mx (H.I.-M.)
2 Instituto de Astrofísica de Andalucía, E-18008 Granada, Spain
3 Departamento de Astrofísica, Universidad de La Laguna, E-38206 La Laguna, Spain
* Correspondence: erika@astro.unam.mx; Tel.: +52-5556223986

**Abstract:** In this work, we present multi-epoch optical spectra of the Seyfert 1.9 galaxy Mrk 883. Data were obtained with the Gran Telescopio Canarias and the *MEGARA* Integral Field Unit mode, archival data from the SDSS-IV MaNGA Survey and the SDSS-I Legacy Survey, and new spectroscopic observations obtained at San Pedro Mártir Observatory. We report the appearance of the broad component of H$\beta$ emission line, showing a maximum FWHM $\sim 5927 \pm 481 \, \mathrm{km \, s^{-1}}$ in the MaNGA spectra, finding evidence for a change from Seyfert 1.9 (23 June 2003) to Seyfert 1.8 (18 May 2018). The observed changing-look variation from Sy1.9 to Sy1.8 has a timescale $\Delta t \sim 15$ y. In addition, we observe profile and flux broad emission line variability from 2018 to 2023, and a wind component in [OIII]5007 Å, with a maximum FWHM $= 1758 \pm 178 \, \mathrm{km \, s^{-1}}$, detected on 15 April 2023. In all epochs, variability of the broad lines was found to be disconnected from the optical continuum emission, which shows little or no variations. These results suggest that an ionized-driven wind in the polar direction could be a possible scenario to explain the observed changing-look variations.

**Keywords:** active galactic nuclei; Seyfert galaxies: variability; optical spectroscopy; AGN winds





## 1. Introduction

The so-called unified model for AGNs [1] has been for a long time the paradigm used to explain the observed phenomenology of the different classes of the jetted and non-jetted AGN family [2]. The model is mainly based on the assumption that a ubiquitous clumpy parsec-scale dusty torus presenting different viewing angles can explain the division into Type 1 (unobscured) and Type 2 (obscured) AGNs (for a review, see [3]).

AGNs are also known for showing multi-wavelength variability with time scales from hours to years (e.g., [4]). Some AGNs display more dramatic variations, including spectral-type variations, e.g., changing from Type 1 to Type 2 classes in time-scales ∼years. AGNs that display this type of variation are identified as 'Changing-look' (CL). They have been observed in the optical and the X-ray bands, and ∼100 CL AGNs have been identified by multi-epoch spectroscopy [5]. CL AGNs are known to present a temporary appearance/disappearance of the broad emission line (BLR) components with timescales from years to decades, and these kinds of coherent changes are found to be difficult to explain—see, e.g., [6,7]. The spectral change variations can be from Type 1 to Type 2, progressing also through the intermediate types, e.g., from Type 1 to 1.8, and 1.9.

The Seyfert galaxy Mrk 883 is a nearby AGN at z = 0.038. Classified as Sy1.9 by [8], it was identified by [9] as an object that shows spectral-type variations. In particular, broad emission line variations were detected in October 1993, producing a change from Sy1.9 to Sy1.8 (or even Sy1.5), with little change in the optical continuum emission. The author demonstrates that the detected line flux changes may not be caused by changes in the

reddening of the BLR, i.e., not due to an obscuring material, suggesting that other mechanisms are involved in the appearance/disappearance of the broad component (BC) in H$\beta$ (H$\beta_{BC}$). Later, analyzing the archival SDSS spectrum of Mrk 883 obtained on 23 June 2003, it was found that the central nucleus has an optical spectrum of Sy1.9 (see [10]). These authors also report that Mrk 883 is a double-peaked narrow emission line AGN (DPAGN), showing one Gaussian component in the locus of Sy galaxies in the Baldwin, Peterson, Terlevich (BPT) diagnostic diagrams [11], and the second Gaussian component is found in the so-called composite region (AGN+SB). This object was also included in a study of X-ray variability of Sy1.8/Sy1.9 galaxies by [12]. Analyzing four observations obtained with XMM–Newton between 2006 and 2010, they reported flux variations of 28% in both the soft and hard X-ray bands with a timescale of $\Delta t \sim 4$ y, and found that Mrk 883 is a Compton-thin (unobscured) AGN.

Mrk 883 also shows disturbed morphology in the optical images of the SDSS [13]. A gaseous ring and tidal tales can be identified, supporting evidence of past merger activity. A double nucleus in its center can be seen—see Figure 1. Our *MEGARA* data presented in this work allowed us to establish that the two nuclear sources (see Section 7), one at the center of Mrk 883 and the other one at the southwest, have a projected separation of 2.2″ or $\sim$1.7 kpc.

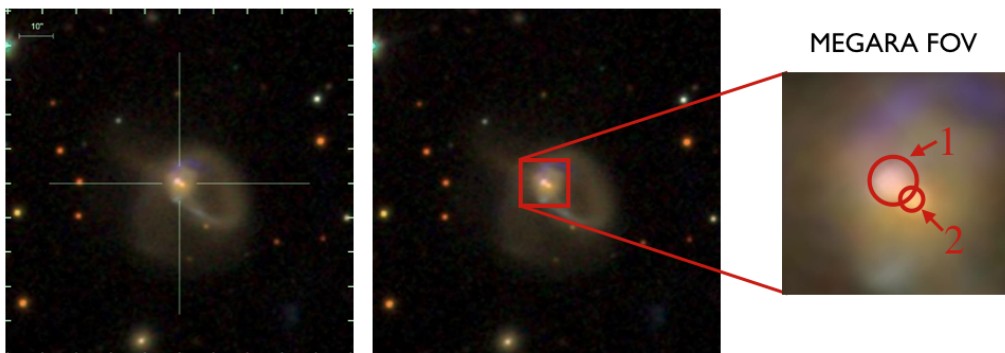

**Figure 1.** (**Left panel**): SDSS RGB color composed image (g-r-i bands), centered at Mrk 883, obtained at MJD 52813 (23 June 2003) showing the two nuclear sources. Mrk 883 is located at the central coordinates. A second nucleus is located in the southwest. (**Central panel**): The same as in the left image, but showing in a red box the *MEGARA* FOV. (**Right panel**): In the *MEGARA* FOV image, the red circles show the two apertures used in this work. Aperture 1 marks the locus of the aperture with a size of 3″. The selection matches the same aperture used in the SDSS fiber aperture position. A second aperture of 1.5″ was selected at the position of the second nucleus.

In this work, we present the results of a new variability study of Mrk 883, through the analysis of optical spectroscopic multi-epoch data obtained with the Gran Telescopio Canarias (GTC) and *MEGARA* in IFU mode, and also analyze the archived 3D spectrum of the central source obtained with the MaNGA survey. We also include the analysis of two long-slit spectra, one from the SDSS database, and a recent spectrum obtained at the Observatorio Astronómico Nacional at San Pedro Mártir (OAN-SPM) in Baja California Mexico, with the 2.1 m telescope and the Boller & Chivens spectrograph. We find that the *MEGARA* spectrum clearly shows the appearance of a broad H$\beta$ emission line component with extended wings. Therefore, we report Seyfert 1.9 to Seyfert 1.8 or CL variability in Mrk 883 on timescales of $\sim$15 years. The variations in the broad components occurred with little changes in the optical continuum emission. We discuss later a possible mechanism that could be triggering the CL variations observed at different epochs. The cosmology adopted in this work is $H_0 = 69.6$ km/s$^{-1}$ Mpc$^{-1}$, $\Omega_m = 0.286$, and $\Omega_\lambda = 0.714$ [14]. Therefore, 1″ corresponds to 756 pc.

## 2. Data Acquisition and Observations

We analyzed the Integral Field Spectroscopy (IFS) data obtained in the Mapping Nearby Galaxies at the Apache Point Observatory survey (MaNGA [15]). Also, we analyzed the single-fiber spectra obtained by the Sloan Digital Sky Survey Data Release 7 (SDSS [13,16,17]). The IFS MaNGA survey aims to obtain systematic observations of 11,273 targets and is part of the SDSS phase IV [18]. MaNGA observations are carried out using five IFUs: 19, 37, 61, 91, and 127 fiber IFU bundles, with a 2″ fiber diameter size [19]. For MaNGA data, the eBOSS spectrograph has a spectral coverage range of 3600 Å to 10,300 Å, and a spectral resolution of R ≈ 2000 [20].

MaNGA observations of MrK 883 were obtained with the 61-fiber IFU on 18 May 2018 (MJD 58256). The single-fiber spectrum of MrK 883 was drawn from the SDSS database. The observation was obtained on 23 June 2003 (MJD 52813). The SDSS uses the Sloan spectrograph, which is the previous version of the MaNGA eBOSS spectrograph with spectral coverage from 3800 Å to 9200 Å and a spectral resolution of R ≈ 2000 [20]. The SDSS single-fiber observation uses a fiber diameter size of 3″.

Additionally, we obtained new IFS observations of Mrk 883 taken with the Multi-Espectrógrafo de Alta Resolución para Astronomía *MEGARA* [21]), located at the Gran Telescopio Canarias (GTC). The observations were obtained as a part of a project dedicated to studying DPAGN at kpc scales to establish the extension and origin of the ionized gas, along with its kinematics (GTC3-21AMEX; PI Benítez). We used the Large Compact Bundle (LCB) mode of *MEGARA* that provides an IFS observation with a field-of-view (FOV) of 12.5″ × 11.3″. We use the Volume Phase Holographic (VPH) gratings VPH675 LR-R and VPH521 MR-G. The VPH675 provides a spectral resolution of R = 5900 and a spectral range of 6096–7303 Å. The VPH521 has a spectral range of 4963–5445 Å, with a spectral resolution R = 12,000. The observations were performed in two observing blocks (OB). The first OB was obtained on 30 March 2021, during a photometric night with a bright moon and an atmospheric seeing of 1.3″. During this OB, three exposures of 663 s for each VPH were carried out. The second OB was obtained on 3 April 2021 on a photometric night with a bright moon and an atmospheric seeing of 1.0″. In this OB, three exposures of 663 s each with the two VPHs were obtained.

Recently, new spectra of Mrk 883 were obtained using the 2.1 m telescope and the Boller & Chivens spectrograph at OAN-SPM (hereafter, these data are denoted as SPM). SPM data were obtained from a long-term follow-up AGN variability program (PI Negrete). Long-slit spectroscopy observations were performed using 600 lines/mm grating and with a slit width of 150 microns. Three exposures of 1800 s were obtained to achieve a minimum signal-to-noise ratio (S/N) ∼ 30. We use two configurations: the blue arm, which covers the spectral range of 3300–5600 Å with a spectral resolution of R = 1000, and the red arm, which covers the spectral range of 5100–7500 Å with a spectral resolution of R = 1300. The observations were taken during gray nights and with photometric conditions on 14 and 17 April 2023.

## 3. Data Reduction

The SPM spectra were reduced using the standard long-slit IRAF packages. Therefore, the final spectra were processed through bias and flat field corrections, and then wavelength calibration, aperture extractions, and, finally, flux calibration.

For the 3D spectra, the reduction procedure is based on the one described in [22]. The objective is to reduce and reconstruct the calibrated *MEGARA* data cube. The procedure is summarized as follows:

(a) 2D reduction: We use the official *MEGARA* Data Reduction Pipeline (DRP) to perform the basic spectroscopic reduction. The DRP executes the bias subtraction, fiber extraction, flat-fielding, fiber flexure correction, and the wavelength and flux calibration of the GTC raw data. In this stage, we correct the spectra for the heliocentric velocity;

(b) 3D reduction: With the extracted wavelength and flux-calibrated multi-fiber spectra, we reconstruct the data cube. We use the 3D spatial re-sampling methodology de-

scribed in [22]. The spatial re-sampling takes into account the *MEGARA* fiber size of 0.6″ to define a data cube of 38 × 40 spaxels with a size of 0.35″. In this stage, we correct the spatial position of the target from the atmospherically differential refraction. In addition, we combine the Observed Blocks (OBs) of the VPH521 MR-G and the VPH675 LR-R. We homogenize the spectral resolution to the lowest VPH resolution: R = 5900, that is, the resolution of VPH675. The final spectral sampling was fixed to a linear sampling of 0.3 Å per pixel. We homogenize the spectral resolution to R = 5900 to apply a full spectral fitting in Section 7;

(c) We perform a second-order spectrophotometric calibration to correct for any flux fluctuation on the surface brightness of the target. The method consists of comparing a set of photometric observations to the synthetic photometry maps from our data cube and measuring any flux deviations. However, the spectral coverage of the VPH521 and VPH675 did not match the full spectral coverage of the standard broadband photometric filters. Therefore, we use the MaNGA IFS observation of Mrk 883 to obtain synthetic photometric maps within the spectral regions of the VPH521 and VPH675. With these maps, we calculate a second-order flux correction map, and use it to obtain the final flux-corrected data cube;

(d) With the final data cube, we now proceed to calculate accurate astrometry. We use both synthetic SDSS r-band images from the MaNGA [15,19] and our final *MEGARA* data cube to match the astrometry and also match it with the SDSS spectrum. To do so, we calculate the centroids of the two nuclei within the MaNGA image and compare them with the centroids of the same nuclei within the *MEGARA* FOV. Therefore, we obtained the World Coordinate System (WCS) for our final *MEGARA* data cube to match the WCS of MaNGA.

## 4. Methodology

We extracted the integrated spectra of the *MEGARA* and MaNGA data cubes using the same aperture of the SDSS fiber to match the archival spectrum of Mrk 883 obtained by the SDSS in MJD 53226. The same aperture was chosen for extracting the spectra of the SPM observations.

The position and size of the SDSS fiber aperture appear in Table 1, and the same for MaNGA, *MEGARA* and SPM data. Also, in Figure 2, we present the emission line profiles in the H$\beta$ (left) and H$\alpha$ (right) regions obtained from *MEGARA*, MaNGA, SPM, and SDSS observations. Using the continuum around 5100 Å, we have estimated S/N ratios of 71, 100, 47, and 41, respectively. We applied the redshift correction using the value provided by the SDSS (z ∼ 0.04).

Similarly, we extracted the spectra for the second aperture for the *MEGARA* data (see Section 7, where we compared both Apertures 1 and 2). The S/N of the continuum around 5100 Å is 47.

**Table 1.** Astrometric position of the SDSS fiber: RA and DEC in (J2000). The aperture radius is in arcsec.

| Instrument | RA | DEC | Aperture Radius | Date | S/N |
|:---:|:---:|:---:|:---:|:---:|:---:|
| SDSS | 16:29:52.886 | +24:26:38.40 | 1.5 | 23 June 2003 | 41 |
| MaNGA | 16:29:52.886 | +24:26:38.40 | 1.5 | 18 May 2018 | 100 |
| *MEGARA* | 16:29:52.886 | +24:26:38.40 | 1.5 | 1 Jan 2021 | 71 |
| *MEGARA* | 16:29:52.794 | +24:26:37.25 | 0.75 | 1 Jan 2021 | 47 |
| SPM | 16:29:52.886 | +24:26:38.40 | 1.5 | 15 April 2023 | 47 |

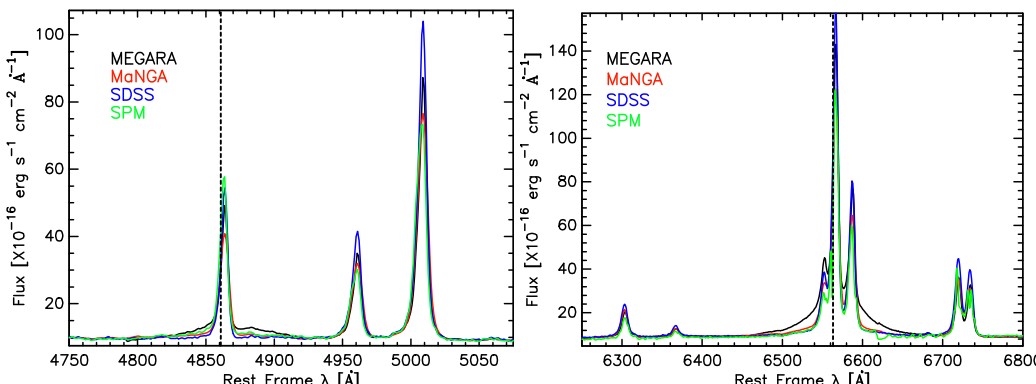

**Figure 2.** (**Left**): Mrk 883 spectra obtained with SPM (green), SDSS (blue), MaNGA (red), and *MEGARA* (black) are shown in the Hβ and [OIII]4959,5007 regions. (**Right**): The same but in the Hα, [NII]λλ6548,6583, and [OI]λλ6300,6364 regions. The abscissa is the rest-frame wavelength. The ordinate is the flux. The dashed vertical lines are in the Hβ and Hα rest-frame positions, respectively.

The best way to quantify the amount of spectral variability in the same object that has multi-epoch spectra is by making the decomposition of the most prominent emission lines. For this purpose, we used the task *Specfit*, which is part of the IRAF astronomical package [23]. *Specfit* allow us to model, at the same time, the underlying continuum, the emission lines, and, if necessary, absorption lines. For the line fitting, we split the spectral range into two regions: 4850 Å and 5050 Å for Hβ, and 6200 Å and 6420 Å for Hα. The Hα region includes the emissions of [NII]λλ6548,6583, [SII]λλ6716,6731, and [OI]λλ6300,6364 doublets. The [OIII]λλ4959,5007 emission is included in the Hβ region. The first step is to set the underlying continuum, modeled using a power law. The emission at the locus of nucleus 1 is dominated by the AGN and completely overshadows the host continuum (see Section 7). We found that the power law index was practically zero for the four spectra. Then, we considered Gaussian profiles to model all the emission line components. Each Gaussian uses three input parameters: the central wavelength, the line intensity, and the FWHM. Once we had the initial guesses (number of components and initial parameters), we minimized the $\chi^2$ to obtain the best fit using the Marquardt algorithm for 5–10 iterations.

We started with the line fitting in the Hα spectral region because it has the most prominent lines. For all the spectra, we modeled Hα, the doublets of [NII], [SII], and [OI] narrow components (NC) constraining the FWHM, to be equal for all the lines. We assumed that they originate in the same clouds and therefore share the same kinematics. The intensities of [NII]λ6548 and [OI]λ6364 were linked to [NII]λ6583 and [OI]λ6300, respectively, taking into account the theoretical 3:1 ratio. The lambda shift of the three doublets was also linked to the same. We included an Hβ_{BC} in the fitting of the four spectra. Since the SDSS spectrum was previously modeled [10], we first reproduced their line fitting, confirming that two Gaussian components are needed to account for the narrow emission lines. We then fit the remaining SPM, MaNGA, and *MEGARA* spectra, imposing the same considerations. The spectral fitting of the Hα region is shown in the right panels of Figure 3.

For the Hβ spectral range, we fit the same two narrow components for the [OIII] doublet and the Hβ_{NC}, trying to set very similar initial conditions in the FWHM and shifts, as found in the Hα region. The residuals obtained in the fitting around [OIII]λ4959 indicated that an additional blueshifted Gaussian component is needed to obtain the best fit. Also, we do not find a broad component for Hβ. However, we have to include an Hβ_{BC} for the SPM, *MEGARA*, and MaNGA spectra (see left panels in Figure 3). Thus, these results are in agreement with the previous fittings carried out on the SDSS spectrum by [10].

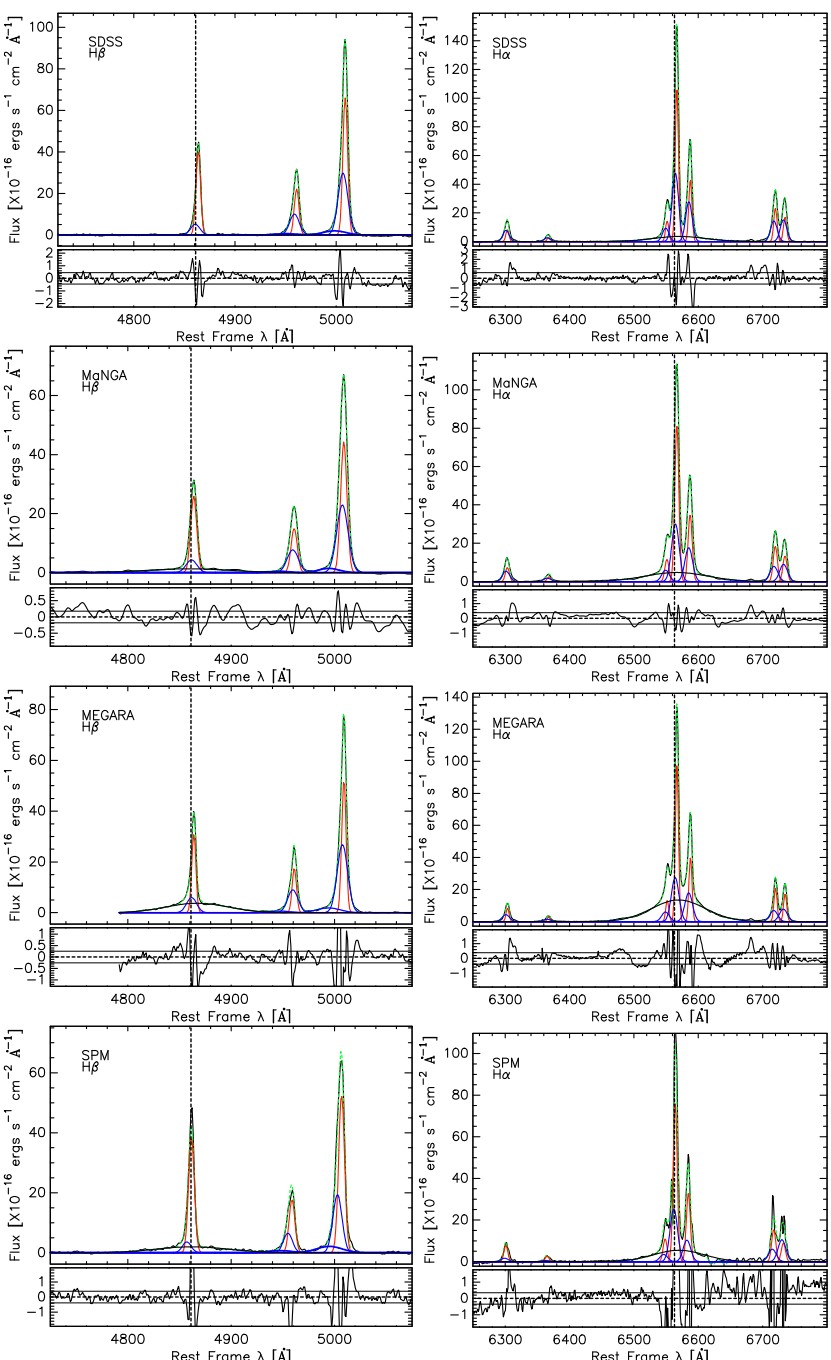

**Figure 3.** (**Left**): Hβ and [OIII] line-fitting continuum subtracted for the SDSS, MaNGA, SPM, and *MEGARA* spectra. (**Right**): The same but in the Hα [NII], [SII], and [OI] regions. The upper panels show the fit, while the lower panels report the residuals. The dashed horizontal black line is at zero level, and the solid lines mark $\pm 2\sigma$. For both spectral ranges, the thin black line is the observed spectra, the green line is the fitted model, and the dashed vertical lines are the Hβ and Hα rest-frames, considering the z reported by the SDSS (z = 0.03787). The thick black line is the Balmer broad component. The red and blue lines are the blue and red components for each narrow emission line. In the Hβ range, the thick blue line is an extra blueshifted component. The abscissa is the rest-frame wavelength. The ordinate is the flux.

## 5. Results

The emission line fluxes, FWHM, and EW(Hα) measurements obtained from the line modeling are presented in Table 2. For the line doublets of [OIII], [NII], and [OI], we

report the fluxes only for the strongest line (see details in Section 4). We report the FHWM only for the Balmer components. The blue and red FWHMs of the forbidden lines are the same for $H\beta_{NC}$ and $H\alpha_{NC}$ components, except for the extra blueshifted component needed to fit the [OIII] 5007 Å. The EWs of $H\alpha_{NC}$ are reported at the bottom of the table. We find consistency in the velocity separation of the blue and red narrow components, with average values of $\Delta v(H\beta)$ $119 \pm 8 \, \mathrm{km\,s^{-1}}$ and $\Delta v(H\alpha)$ $107 \pm 10 \, \mathrm{km\,s^{-1}}$, in agreement with [10] for the SDSS spectrum. The average shift velocities with respect to the rest frame are $109 \pm 12 \, \mathrm{km\,s^{-1}}$, $113 \pm 12 \, \mathrm{km\,s^{-1}}$, and $-45 \pm 8 \, \mathrm{km\,s^{-1}}$ for the broad, red, and blue components, and $-689 \pm 5 \, \mathrm{km\,s^{-1}}$ extra blueshifted component for [OIII]. This component is denoted as the wind component.

**Table 2.** Mrk 883 line fluxes in units of $10^{-16} \, \mathrm{erg\,s^{-1}\,cm^{-2}\,Å^{-1}}$, FWHM of the broad and narrow components in $\mathrm{km\,s^{-1}}$, and $H\alpha_{NC}$ EW in Å.

| Line | SDSS | MaNGA | *MEGARA* | SPM |
|---|---|---|---|---|
| **Flux** | | | | |
| $H\alpha_{BC}$ | 355.2 ± 10.7 | 570.7 ± 8.2 | 1406.4 ± 13.3 | 534.0 ± 29.6 |
| $H\alpha_{NC}$ B | 600.0 ± 6.0 | 446.4 ± 20.1 | 410.0 ± 7.8 | 370.1 ± 56.6 |
| $H\alpha_{NC}$ R | 671.6 ± 5.7 | 652.4 ± 20.1 | 594.6 ± 7.3 | 599.0 ± 11.4 |
| [NII]$\lambda6583$ B | 350.0 ± 7.0 | 263.1 ± 10.5 | 268.6 ± 7.8 | 151.8 ± 20.4 |
| [NII]$\lambda6583$ R | 272.2 ± 5.8 | 281.7 ± 10.1 | 243.5 ± 5.6 | 258.7 ± 8.5 |
| [SII]$\lambda6716$ B | 183.9 ± 5.2 | 120.3 ± 6.4 | 106.2 ± 4.5 | 90.0 ± 13.4 |
| [SII]$\lambda6716$ R | 150.1 ± 4.3 | 151.1 ± 5.6 | 129.2 ± 3.5 | 123.6 ± 12.5 |
| [SII]$\lambda6731$ B | 194.3 ± 5.3 | 136.3 ± 5.9 | 119.8 ± 4.6 | 158.4 ± 15.2 |
| [SII]$\lambda6731$ R | 109.1 ± 4.4 | 109.5 ± 5.5 | 107.3 ± 3.6 | 72.1 ± 17.0 |
| [OI]$\lambda6300$ B | 94.3 ± 3.8 | 77.3 ± 4.1 | 62.1 ± 3.4 | 23.4 ± 10.0 |
| [OI]$\lambda6300$ R | 49.7 ± 2.9 | 56.6 ± 3.6 | 48.5 ± 2.5 | 59.0 ± 7.1 |
| $H\beta_{BC}$ | . . . | 134.2 ± 12.3 | 281.4 ± 8.8 | 174.1 ± 16.3 |
| $H\beta_{NC}$ B | 59.5 ± 19.8 | 57.6 ± 5.6 | 73.3 ± 5.7 | 38.6 ± 6.5 |
| $H\beta_{NC}$ R | 231.2 ± 4.9 | 193.3 ± 5.1 | 157.6 ± 4.9 | 270.0 ± 10.0 |
| [OIII]$\lambda5007$ B | 357.2 ± 5.1 | 317.2 ± 12.1 | 340.0 ± 12.1 | 214.0 ± 10.0 |
| [OIII]$\lambda5007$ R | 390.1 ± 0.8 | 340.3 ± 13.5 | 276.4 ± 13.3 | 380.0 ± 10.0 |
| [OIII]$\lambda5007_W$ | 56.0 ± 19.8 | 36.8 ± 5.6 | 53.9 ± 5.7 | 66.0 ± 6.5 |
| **FWHM** | | | | |
| $H\alpha_{BC}$ | 4336 ± 172 | 5181 ± 112 | 4473 ± 88 | 4199 ± 121 |
| $H\alpha_{NC}$ B | 538 ± 6 | 636 ± 11 | 642 ± 12 | 629 ± 92 |
| $H\alpha_{NC}$ R | 270 ± 2 | 345 ± 4 | 262 ± 2 | 338 ± 7 |
| $H\beta_{BC}$ | . . . | 5927 ± 481 | 4353 ± 117 | 5076 ± 373 |
| $H\beta_{NC}$ B | 672 ± 144 | 777 ± 78 | 712 ± 71 | 621 ± 25 |
| $H\beta_{NC}$ R | 332 ± 21 | 431 ± 6 | 297 ± 6 | 408 ± 6 |
| [OIII]$\lambda5007_W$ | 1586 ± 159 | 1425 ± 110 | 1599 ± 224 | 1758 ± 178 |
| **EW** | | | | |
| $H\alpha_{NC}$ B | 64.49 ± 0.69 | 49.55 ± 1.20 | 45.30 ± 0.94 | 43.20 ± 6.67 |
| $H\alpha_{NC}$ R | 72.18 ± 0.66 | 72.41 ± 2.29 | 65.70 ± 0.88 | 69.92 ± 6.98 |
| **Derived quantities** | | | | |
| $\log L_{bol}$ | 44.23 ± 0.20 | 44.25 ± 0.19 | 44.26 ± 0.18 | 44.24 ± 0.20 |
| $\log M_{BH}$ | 7.93 ± 0.20 | 8.07 ± 0.19 | 7.92 ± 0.20 | 7.80 ± 0.20 |
| $\log L_{Edd}$ | 46.03 ± 0.20 | 46.17 ± 0.19 | 46.03 ± 0.08 | 45.90 ± 0.20 |
| $\log L_{bol}/L_{Edd}$ | 0.02 ± 0.01 | 0.01 ± 0.01 | 0.02 ± 0.01 | 0.02 ± 0.02 |
| Balmer decrement | . . . | 4.3 ± 0.5 | 5.0 ± 0.2 | 3.1 ± 0.5 |

For each spectrum, we computed the bolometric luminosity ($L_{bol}$) using the continuum luminosity at 5100 Å and considering a bolometric correction of 10.33 [24], the black hole mass ($M_{BH}$, following [25] using the FWHM($H\beta_{BC}$)), the Eddington luminosity ($L_{Edd}$), and the Eddington ratio ($L_{bol}/L_{Edd}$), defined as the ratio of the bolometric and Eddington luminosities. For the SDSS spectrum, we calculated the FWHM($H\beta_{BC}$) using Equation (3) in [26]. We find average values using the four epochs of $\log L_{bol} = 44.25 \pm 0.19$, $\log M_{BH} \simeq 7.93 \pm 0.20$, $\log L_{Edd} = 46.03 \pm 0.17$, and $\log L_{bol}/L_{Edd} = 0.02 \pm 0.02$, with a standard deviation of 0.01, 0.11,

0.01, and 0.005, respectively. Note that, in Sy1.8 and Sy1.9, black hole mass estimates can be affected by possible obscuration of the continuum and the BLR. We report the estimations of the Balmer decrement (BD) and find that it also shows variability. This is an expected result in AGN with a low Eddington ratio (see Figure 2 in [27]).

*Diagnostic Diagrams*

The emission line profile modeling, previously shown, allows us to explore the ionized emission properties of Mrk 883 at different epochs. For instance, the line ratios were used to obtain BPT diagnostic diagrams. The regions in the BPT diagram are delimited by the [28] demarcation line (red line in Figure 4): above that line, the objects have an ionization that can be associated with an AGN, and the [29] demarcation line (black line in Figure 4): below that region, the ionization can be associated to an SF region. Objects that fall in between SF and AGN regions are usually considered composite. In this region, we can have objects with emissions produced by aged stars that are difficult to identify. As such, these objects can be misidentified from being aged stars as a LINER-like object [30]. The area below the [31] line denotes the LINER region (blue line in Figure 4). Therefore, we have also obtained the EWH$\alpha$ vs. [NII]/H$\alpha$ diagram (WHAN [30]) using the line ratios obtained in this work. The WHAN diagram is designed to disentangle the ionization source produced by Hot Low-mass evolved stars (HOLMES [30,32]) from AGNs and SF regions. [30] separate the WHAN diagram into four regions: the SF region— log [NII]/H$\alpha$ < 0.4 and EW(H$\alpha$) > 3 Å; the fake AGN region—EW(H$\alpha$) < 3 Å; the weak AGN region—log [NII]/H$\alpha$ > −0.4 and 3 Å < EW(H$\alpha$) < 6 Å; and the strong AGN emission— log [NII]/H$\alpha$ > −0.4 and EW(H$\alpha$) > 6 Å. The BPT diagrams and the WHAN diagram obtained for Mrk 883 are shown in Figure 4.

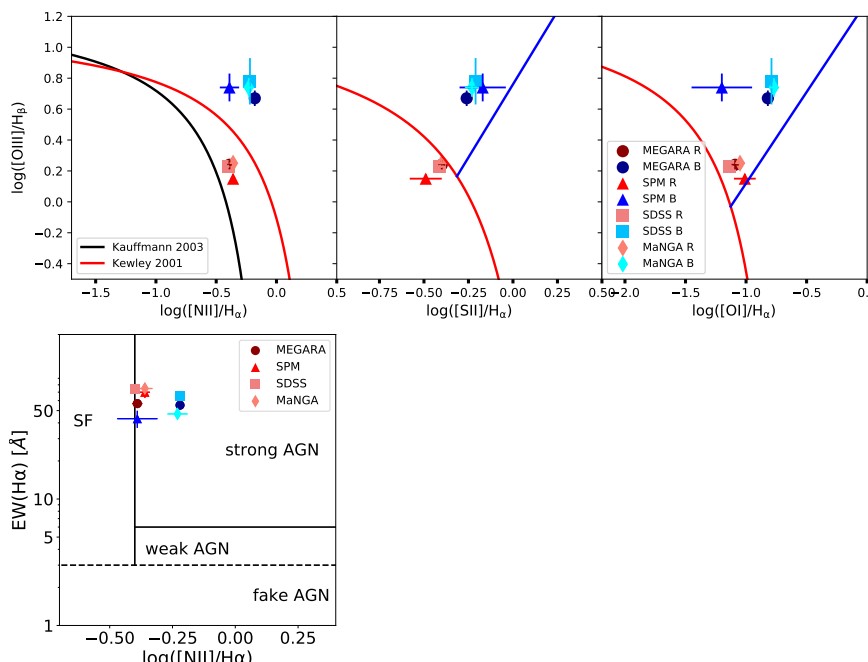

**Figure 4. (Upper panels)**: BPT diagrams [11] obtained with the line ratios of Mrk 883. Different colors and symbols were used to illustrate the two Gaussian components needed to fit the narrow emission lines obtained during the multi-epoch observations. The line ratios obtained with the red Gaussian components from SDSS, MaNGA, *MEGARA*, and SPM spectra appear in the locus of composite objects in two diagrams and as AGN in the third one. Note that the *MEGARA* points are behind the pink SDSS and MaNGA points. In the case of the blue Gaussian components, these appear in the AGN locus. For more details, see the text. **(Lower panel)**: The WHAN diagram (same colors and symbols used in the upper panels) shows that the blue and red Gaussian components from SDSS, MaNGA, *MEGARA*, and SPM appear, within the errors, in the strong AGN locus.



## 6. Variability Analysis

Flux variations in the Balmer broad components were detected, using the SDSS spectrum as a reference, in the MaNGA, *MEGARA*, and SPM spectra. Using the spectral decomposition described in Section 4, we have isolated the H$\alpha$ and H$\beta$ broad emission lines to compare the four activity states. This was performed through the subtraction of the modeled narrow components. In the upper panels of Figure 5, we show the profile variations in H$\beta_{BC}$ and H$\alpha_{BC}$ in the four epochs (with the NLR subtracted). The minimum intensity and width of broad lines were detected in the SDSS spectrum, and the maximum flux was detected in the *MEGARA* spectrum. However, it is worth noting that the maximum FWHM for the H$\alpha$ and H$\beta$ broad components were found in MaNGA data. Also, we found that the MaNGA and SPM have activity states in between the maximum and minimum activity states.

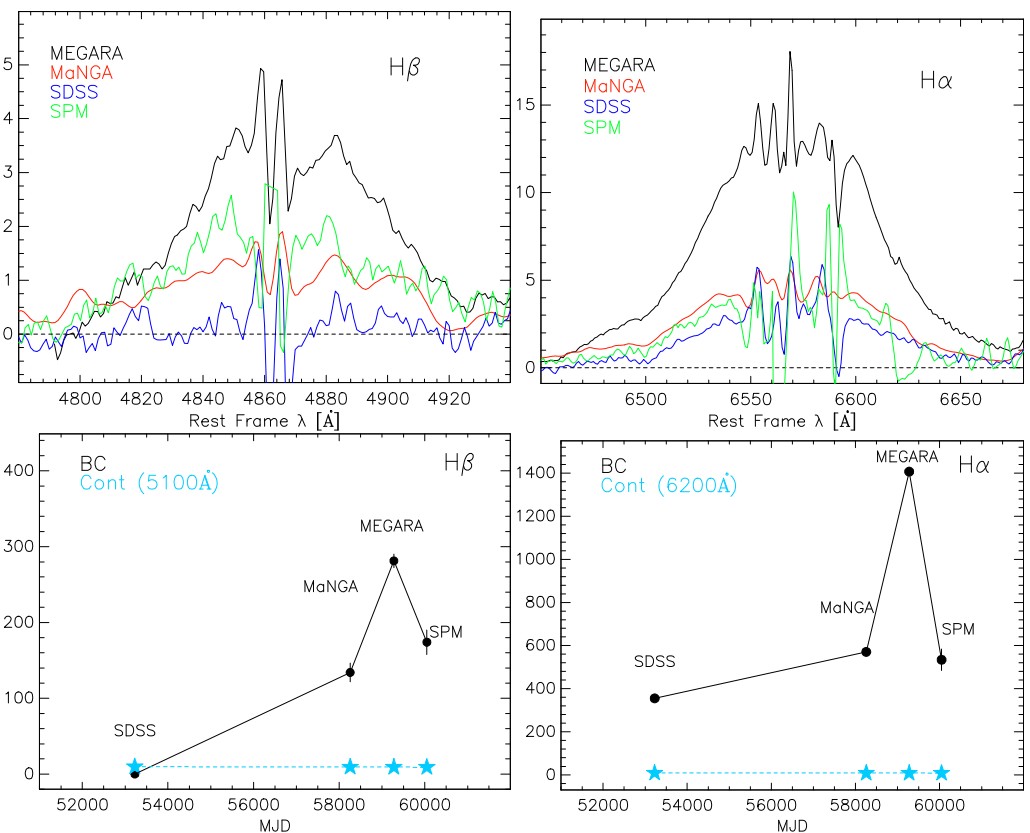

**Figure 5.** (**Upper panels**): Profile variations in H$\beta_{BC}$ and H$\alpha_{BC}$ are observed in four different epochs. The SDSS spectrum shows no broad H$\beta$ component. The horizontal axis shows the MJD. The vertical shows the flux in units $10^{-16}$ erg s$^{-1}$ cm$^{-1}$ Å$^{-1}$. The appearance of the H$\beta_{BC}$ is evident in the MaNGA spectrum. The maximum intensity of H$\beta_{BC}$ and H$\alpha_{BC}$ was observed three years later in the *MEGARA* spectrum. The SPM spectrum shows that the intensity of H$\beta_{BC}$ and H$\alpha_{BC}$ is declining after its maximum. (**Lower panels**): The optical continuum flux measured at 5000 Å and in 6200 Å (turquoise-filled stars) shows little or no variation during the four epochs, while flux variations (same units as in upper panels) are evident in the H$\beta$ and H$\alpha$ broad components.

A more detailed variability analysis is shown in the lower panels of Figure 5. The flux variations for the four stages of the H$\alpha$ and H$\beta$ broad components as a function of time are presented. The lower left panel shows the flux variations in H$\beta$ and the flux continuum values for each epoch, measured around 5100 Å using a window of 50 Å. It is important to mention that the continuum flux at 5100 Å remains almost constant at around $9.1 \pm 0.2 \times 10^{-16}$ erg s$^{-1}$ cm$^{-2}$ Å$^{-1}$. Similar variations are seen in H$\alpha$ and the flux continuum around 6200 Å measured using a window of 50 AA, shown in the lower right

panel. For both Hα and Hβ broad components, we can observe clear flux variations, where *MEGARA* spectrum shows the maximum flux values in the Hα$_{BC}$ and Hβ$_{BC}$.

## 7. Spatially Resolved *MEGARA* Data

Until this point, we have discussed only the data analysis of the nucleus of Mrk 883 (Aperture 1). Now, we will explore the spatially resolved broad emission line components of Hβ and Hα spectral regions in both nuclei (see Figure 1). We model the emission line profiles spaxel by spaxel using BADASS3D (Bayesian AGN Decomposition Analysis for SDSS Spectra, [33]) software tool. BADASS3D allow us to perform a Bayesian spectral decomposition of the host galaxy contribution and a multi-component emission line fit of the emission lines at the same time for IFS data. With this tool, we performed a modeling of the broad components of the Hα and Hβ spectral regions plus the host galaxy for each of the 38 × 40 spaxels that comprise the *MEGARA* FOV. To minimize the host-AGN degeneracy on the spectral fitting, we applied BADASS3D to fit the full spectral range from 4963 to 7300 once, masking the gap between the two VPHs. Therefore, it was necessary to homogenize the spectral resolution of R = 5900. The emission line modeling is not as detailed as the SPECFIT modeling that we performed in the previous section, but allows us to obtain the automatic modeling of the spectral lines along all the FOV of *MEGARA* with good accuracy (see [33] for more details). Therefore, we show the line modeling results from both nuclei (see Figure 6).

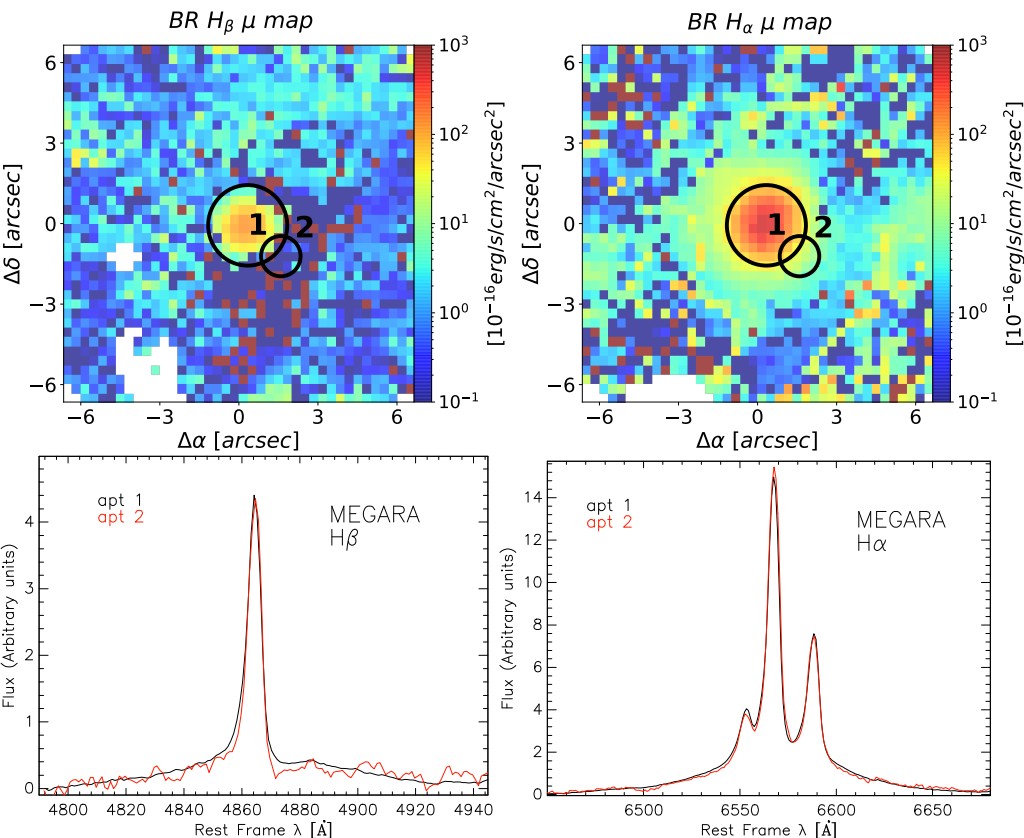

**Figure 6.** (**Upper panel**): Spatially resolved images showing the total flux of the broad Hβ and Hα components. (**Lower panel**) Nucleus 2 (Aperture 2) spectrum was analyzed with the code BADASS3. Since both nuclei are separated by ∼1.7 kpc and nucleus 2 shows a very similar spectrum to nucleus 1, the analyses provide similar results. This is due to ionized gas originating from the AGN in Mrk 883 that contaminates (at kpc scales) the emission from nucleus 2.

In Figure 6, we show the locus occupied by both nuclei. The *MEGARA* data were used to estimate the projected separation in the sky between both nuclei (∼1.7 kpc) and we also found that the spatial distribution of the flux of the broad component is always

located at nucleus 1 (Aperture 1). Our data did not show that nucleus 2 is producing AGN activity. Instead, the spatial analysis shows that the emission line spectrum of this source is the result of ionized gas produced by nucleus 1 that is spread by the PSF to nucleus 2. This explains why the integrated spectra from aperture 2 show a very similar spectrum and, therefore, this spectrum was not analyzed. The contamination of the spectra from nuclei 1 to 2 does not allow for disentangling the true nature of nucleus 2.

## 8. Discussion

The multi-epoch spectral observations show that flux variations are related to spectroscopic changes. Since the $H\beta_{BC}$ is not observed in the SDSS spectrum, we found, in agreement with previous results by [10], that the SDSS spectrum is consistent with a Sy1.9 galaxy. Fifteen years later, we observed the appearance of the $H\beta_{BC}$ in the MaNGA spectra, consistent with a Sy1.8 galaxy. Therefore, our multi-epoch observations show that Mrk 883 is a CL AGN. Since the CL behavior was previously observed by [9], where he reported spectral-type variations from Sy1.9 (May 1979) to Sy1.8 (October 1993), we have found an object that presents recurrent CL variability in timescales ∼15 years in the optical bands. Since we do not have data between SDSS (2003) and MaNGA (2018) spectra, the recurrent variability timescale could be shorter. This could be better established if yearly optical spectral monitoring of the source is performed. Our multi-epoch spectra allow us to observe the evolution of the Balmer line profiles during the last five years. The variations include profile and flux variations, with an almost non-varying optical continuum.

Since the observed optical continuum variation in Mrk 883 is small, the appearance of the broad component in $H\beta$ may not be caused by changes in the accretion rate, i.e., changes may be due to obscuration by outflows. Some models suggest even accelerating outflows [6,34,35] as the origin of the CL variations. Accretion disk instabilities can also explain the CL variations, e.g., [36,37]. For example, radiation pressure instabilities can provide a viable mechanism to explain repeated outbursts on timescales of a few years (see [38]). Some studies have also found that CL AGNs tend to have low Eddington ratios, e.g., [37]. Our data confirm that Mrk 883 has low Eddington ratio values. For this kind of CL AGN, the disk-wind broad-line-region model is found to be a plausible explanation of the CL phenomenon [5,37]. Recently, [7] suggested that CL AGNs could be produced by non-steady outflows even when the luminosity of the central source remains constant throughout variations. Thus, dynamical processes occurring outside the accretion disk can be the origin of the variations. Therefore, based on our observations, it could be possible that an ionized-driven wind originating outside the accretion disk escaping in the polar direction could be producing the CL behavior in Mrk 883.

In addition, we report, for the first time, the presence of a wind component with a maximum FWHM = $1758 \pm 178$ km s$^{-1}$ in the [OIII]5007 Å line in Mrk 883. It is worth noting that the wind component is narrower when the flux and width of the broad components increase. Also, we find that, in the spatially resolved WHAN diagram, Mrk 883 is a strong AGN. On the other hand, due to the contamination produced by the closeness of the two nuclei (∼1.7 kpc), we cannot establish the true nature of the second one. The study of nature of the second nuclei and the origin of the wind component will be addressed in a forthcoming paper.

**Author Contributions:** Conceptualization, E.B. and C.A.N.; methodology, E.B., H.I.-M. and C.A.N.; software, C.A.N. and H.I.-M.; validation, I.C.-G. and J.M.R.-E.; formal analysis, E.B., H.I.-M. and C.A.N.; investigation, E.B.; resources, E.B., H.I.-M., C.A.N., I.C.-G. and J.M.R.-E.; data curation, H.I.-M.; writing—original draft preparation, E.B., H.I.-M. and C.A.N.; writing—review and editing, E.B., H.I.-M. and I.C.-G.; visualization, H.I.-M.; supervision, E.B.; project administration, I.C.-G.; funding acquisition, I.C.-G. All authors have read and agreed to the published version of the manuscript.

**Funding:** All authors acknowledge support from DGAPA-UNAM grant IN119123. E.B. and I.C.-G. thank the support from CONAHCyT grant CF-2023-G-100. C.A.N. thanks support from CONAHCyT project Paradigmas y Controversias de la Ciencia 2022-320020. C.A.N. and H.I.-M. thank support from DGAPA-UNAM grant IN111422. H.I.-M. thanks support from DGAPA-UNAM grant IN106823.

J.M.R.-E. acknowledges the Spanish State Research Agency under grant number AYA2017-84061-P. J.M.R.-E. also acknowledges the Canarian Government under the project PROID2021010077, and is indebted to the Severo Ochoa Programme at the IAC. Funding for the SDSS and SDSS-II has been provided by the Alfred P. Sloan Foundation, the Participating Institutions, the National Science Foundation, the U.S. Department of Energy, the National Aeronautics and Space Administration, the Japanese Monbukagakusho, the Max Planck Society, and the Higher Education Funding Council for England. The SDSS website is www.sdss4.org. Funding for the Sloan Digital Sky Survey IV has been provided by the Alfred P. Sloan Foundation, the U.S. Department of Energy Office of Science, and the Participating Institutions.

**Data Availability Statement:** The GTC/*MEGARA* data are available in the public archive at https://gtc.sdc.cab.inta-csic.es/gtc/index.jsp (accessed on 21 December 2023). SDSS data are available at https://data.sdss.org/sas/ (accessed on 21 December 2023). The SPM data underlying this article will be shared on reasonable request to the corresponding author.

**Acknowledgments:** This work is based on observations made with the Gran Telescopio Canarias (GTC), installed at the Spanish Observatorio del Roque de los Muchachos of the Instituto de Astrofísica de Canarias, on the island of La Palma. This work is based on data obtained with *MEGARA* instrument, funded by European Regional Development Funds (ERDF), through Programa Operativo Canarias FEDER 2014-2020. SDSS acknowledges support and resources from the Center for High-Performance Computing at the University of Utah. The SDSS website is www.sdss4.org. This work is based on observations at the Observatorio Astronómico Nacional on the Sierra San Pedro Mártir (OAN-SPM), Baja California, Mexico. IRAF was distributed by the National Optical Astronomy Observatory, managed by the Association of Universities for Research in Astronomy (AURA) under a cooperative agreement with the National Science Foundation. This research has made use of the NASA/IPAC Extragalactic Database (NED), which is operated by the Jet Propulsion Laboratory, California Institute of Technology, under contract with the National Aeronautics and Space Administration.

**Conflicts of Interest:** The authors declare no conflicts of interest.

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
