# Peer review of "Multi-Epoch Optical Spectroscopy Variability of the Changing-Look AGN Mrk 883"

_universe, doi:10.3390/universe10010021_

Round 1
Reviewer 1 Report
Comments and Suggestions for Authors
Referee report on the manuscript by Benitez et al.
The authors present an excellent spectroscopic data set on the nearby galaxy Mrk 883, including archival data, and new data they obtained. Through detailed spectral fits they find evidence for variability in the broad Balmer lines, and report a change in the Seyfert sub-type, from Seyfert 1.9 to Seyfert 1.8.
I have a number of comments for the author's consideration.
First, a main comment regards the apparent evidence for variability in the narrow lines on timescales of just years.
This is almost certainly an artefact of the calibration and/or line decomposition techniques.
We do not expect that the NLR of AGNs is variable on the time scale of years. NLRs are widely extended, with light travel times of 100s-1000s of years, and recombination timescales of 1000s of years (low gas density). Therefore, the NLR is not expected to vary on human timescales.
Indeed, variable NLRs are never observed/reported in the literature (with the single exception of dramatic outbursts that can sometimes affect the very inner NLR, called coronal-line region).
Usually, what is done when observing nearby AGN at varying atmospheric seeing and/or imperfect absolute photometry,
is to recalibrate all spectra assuming constand NLR emission, because we know that the NLR cannot vary significantly on short times.
Therefore, I strongly suggest that the authors remove most of their claims of variable narrow lines. One possibility is to mention very briefly, that despite (very good!) attempts to ensure similar aperture sizes and location for the spectral extraction regions, there is always a remaining uncertainty in flux calibration due to variable atmospheric seeing, that can not be completely corrected for, because the pointlike BLR, the extended blobby NLR, and the differently extended host emission are all affected differently.
Mrk883 is a particularly challenging case to analyze because of the complicated geometry and its two nuclei. Of course, the latter also make the galaxy especially interesting to study, inclzding a search for dual AGN.
What I find interesting and noteworthy is the fact, that even after "re-calibrating" for constant NLR emission, the broad lines, esp Halfa, are still required to vary, and so the BLR variability is very likely real. I would make this the main message of the publication, and emphasize, that any apparent NLR variability is considered an artefact (for the reasons given above).
More comments, in the order they appear in the manuscript.
-- Line 28: I suggest to rephrase the sentence: CL AGN are not really inconsistent with the unified model. Already since the 1970s, it has been known that AGN are variable in continuum and BLR, and it has always been accepted, that AGN variability is an extra factor in addition to any strict version of the unified model. The new topic that has been brought up in recent years, are not violations of the unified model, but rather the concern that highly variable CL Seyferts which dramatically change their UV-optical continuum pose challenges to current accretion disk theory (e.g., Lawrence 2018, Nature).
So my suggestion is, to rephrase the sentence for instance like this:
While CL AGN are not inconsistent with unified models (we always knew that AGN are variable), the high amplitudes of continuum variability in the optical and UV regime pose challenges to accretion disk models (e.g., Lawrence 2018).
-- line 34 (also line 84, and at several other spots in the manuscript): ".... with little changes in the continuum emission" --> add the word "optical", so it reads "optical continuum emission".
This just highlights, that we only ever observe the ootical (or UV) continuum, but emission lines are powered by the unobservable EUV continuum, which might well be variable.
Also, in the case of Mrk 883, I wondered, if the optical emission is dominated by the AGN, or rather by the host galaxy ? That would be useful to add, if known.
(see also line 163: no host emission was midelled, just a powerlaw: is this because there is no evidence for host emission ?)
For instance, it could be possible, that the optical AGN continuum is indeed variable, but is always much fainter than the host (strong extinction), and therefore little overall continuum variability is detected.
-- line 44: if known, would be interesting to add here, by how much the X-ray emission varied: like few %, or factor 2, or factor 10, ... ?
-- Fig. 3, Sect. 5: I am sure the authors already checked this, but would be useful to add one sentence: could the non-detection if broad Hbeta in SDSS perhaps be due to the lower spectral quality, i.e., lower S/N ?
-- Fig. 5: would it be quick to re-fit the SDSS spectrum, enforcing a broad Gauss component (FWHM fixed) ? This, in order to get an upper limit on any faint emission that could still be present in the noise.
Then, that upper limit could repkace the number of "0" in Fig. 5.
-- Fig. 5, etc., and related text: As mentioned above, I stronlgy suggest to delete most mentions of variable NLR emission, since unphysical and very likely an artefact.
-- line 203: It is very useful to estimate the SMBH mass of the system. It is rarely done for Sy types 1.8 and 1.9 because a large fraction of the BLR is believed to be obscured, and the continuum as well. So, a cautious comment on this might be given in a footnote. It is still a useful order of magnitude estimate.
It is fully up to the authors, but they could add a second SMBH mass estimate, based on sigma_[OIII] (as surrogate for stellar velocity dispersion), since they have these great optical spectra which cover [OIII] and even separately for both nuclei.
-- Tab. 2: MEGARA, 2nd row entry: has error bars of +-0.0 assigned: a typo ?
-- Tab. 2: It would be interesting to inspect and briefly report the Balmer decrement; i.e., the flux ratio of broad Halfa/Hbeta. Deviations from the recombination value of ~3 could be used to measure extinction along the line of sight. Looks like no strong extinction, but some. But surprisingly and perhaps unexpectedly, the Balmer decrement is highest, when the Balmer lines are brightest....
-- Caption of Fig. 4, last sentence: To me, the red SDSS data point is still within (just at the border, within the errors) of the strong AGN regime. Again, because significant variability of a NLR is not expected on years timescale (but on 100s to 1000s if years), the NLR ionization should remain the same.
-- Sect. 7: Great idea to search for a second AGN in the second nucleus via the BLR emission. And I fully agree with the author's conclusion, that the identical spectra just imply seeing-based scattered emission from the brighter nucleus into the nearby regions.
-- line 301: see my earlier comment, that it is only the *optical continuum* that is observed (to be constant), while the lines are driven by the unobserved EUV continuum, etc.
So I suggest to leave open several possible scenarios: dynamical processes are certainly a possibility, but also alternatively accretion-disk changes affecting the unobservable EUV, or the possibility that the observed optical emission is host dominated (so the AGN continuum remains undetected even in the optical, and then could well be variable intrinsically).
Author Response
The authors present an excellent spectroscopic data set on the nearby galaxy Mrk 883, including archival data, and new data they obtained. Through detailed spectral fits they find evidence for variability in the broad Balmer lines, and report a change in the Seyfert sub-type, from Seyfert 1.9 to Seyfert 1.8.
I have a number of comments for the author's consideration.
First, a main comment regards the apparent evidence for variability in the narrow lines on timescales of just years.
This is almost certainly an artefact of the calibration and/or line decomposition techniques.
We do not expect that the NLR of AGNs is variable on the time scale of years. NLRs are widely extended, with light travel times of 100s-1000s of years, and recombination timescales of 1000s of years (low gas density). Therefore, the NLR is not expected to vary on human timescales.
Indeed, variable NLRs are never observed/reported in the literature (with the single exception of dramatic outbursts that can sometimes affect the very inner NLR, called coronal-line region).
Usually, what is done when observing nearby AGN at varying atmospheric seeing and/or imperfect absolute photometry, is to recalibrate all spectra assuming constand NLR emission, because we know that the NLR cannot vary significantly on short times.
Therefore, I strongly suggest that the authors remove most of their claims of variable narrow lines. One possibility is to mention very briefly, that despite (very good!) attempts to ensure similar aperture sizes and location for the spectral extraction regions, there is always a remaining uncertainty in flux calibration due to variable atmospheric seeing, that can not be completely corrected for, because the pointlike BLR, the extended blobby NLR, and the differently extended host emission are all affected differently.
Mrk883 is a particularly challenging case to analyze because of the complicated geometry and its two nuclei. Of course, the latter also make the galaxy especially interesting to study, inclzding a search for dual AGN.
Agree with the referee that NLR variability has larger time-scales due to its extension, that it is rarely reported in the literature and that the variations found with the data analyzed in the paper are not NLR variations, but artifacts may be due to variable seeing as noted by the referee. We removed the claims as recommended.
Regarding the recalibration, the NLR were modeled in each of the four epochs, and subsequently subtracted. The BLR profile variations are shown in the upper panel of Figure 5, and the SDSS spectrum was considered as a reference to show the profile variability. We have removed all paragraphs/figures related to NLR variations.
What I find interesting and noteworthy is the fact, that even after "re-calibrating" for constant NLR emission, the broad lines, esp Halfa, are still required to vary, and so the BLR variability is very likely real. I would make this the main message of the publication, and emphasize, that any apparent NLR variability is considered an artefact (for the reasons given above).
Agree. To avoid confusion, the NLR variations shown in the middle panel of Fig 5 were removed. A new figure for the middle panel is presented, showing only variations of the broad lines.
Also, in the same figure, the lower panel has been removed.
More comments, in the order they appear in the manuscript.
-- Line 28: I suggest to rephrase the sentence: CL AGN are not really inconsistent with the unified model. Already since the 1970s, it has been known that AGN are variable in continuum and BLR, and it has always been accepted, that AGN variability is an extra factor in addition to any strict version of the unified model. The new topic that has been brought up in recent years, are not violations of the unified model, but rather the concern that highly variable CL Seyferts which dramatically change their UV-optical continuum pose challenges to current accretion disk theory (e.g., Lawrence 2018, Nature).
So my suggestion is, to rephrase the sentence for instance like this:
While CL AGN are not inconsistent with unified models (we always knew that AGN are variable), the high amplitudes of continuum variability in the optical and UV regime pose challenges to accretion disk models (e.g., Lawrence 2018).
We thank the referee for the suggestion, but we have decided to remove the phrase, instead.
-- line 34 (also line 84, and at several other spots in the manuscript): ".... with little changes in the continuum emission" --> add the word "optical", so it reads "optical continuum emission".
This just highlights, that we only ever observe the ootical (or UV) continuum, but emission lines are powered by the unobservable EUV continuum, which might well be variable.
Done.
Also, in the case of Mrk 883, I wondered, if the optical emission is dominated by the AGN, or rather by the host galaxy ? That would be useful to add, if known.
(see also line 163: no host emission was modeled, just a powerlaw: is this because there is no evidence for host emission ?)
In section 7 we performed a full spectral fitting using a bayesian method (we use BADASS3D code) to model the host emission and the AGN contribution. From this analysis, we determine that nuclei 1 emission is dominated by the AGN and the host galaxy emission represents a small contribution of the total flux. We add an explanatory text indicating that the AGN overshadows the host emission at nuclei 1. Therefore, host emission was not modeled.
For instance, it could be possible, that the optical AGN continuum is indeed variable, but is always much fainter than the host (strong extinction), and therefore little overall continuum variability is detected.
In this case, the host galaxy does not dominate the central emission, and our data show that the optical continuum has very small variations. But, as you mentioned before, EUV variations could be present and unobservable in our data.
-- line 44: if known, would be interesting to add here, by how much the X-ray emission varied: like few %, or factor 2, or factor 10, ... ?
The paper by Hernández-García et al. 2017 found X-ray variations in Mrk 883 with timescales of 4 years, this is already mentioned in line 45. We rewrote the paragraph, adding that these authors also found emission variations of 28% in the soft and hard X-ray bands. Data analyzed in that work are the only one available in X-rays.
-- Fig. 3, Sect. 5: I am sure the authors already checked this, but would be useful to add one sentence: could the non-detection if broad Hbeta in SDSS perhaps be due to the lower spectral quality, i.e., lower S/N ?
The SDSS spectrum has S/N~41. The non detection of broad component of Hbeta in this spectrum was published by Benítez et al. 2013. We still have re-modeled this same spectrum, and find the same result, i.e. the spectrum shows no broad Hbeta component.
-- Fig. 5: would it be quick to re-fit the SDSS spectrum, enforcing a broad Gauss component (FWHM fixed) ? This, in order to get an upper limit on any faint emission that could still be present in the noise.
Then, that upper limit could repkace the number of "0" in Fig. 5.
The top left panel of Figure 5 gives a clear view of the absence of the broad component of Hbeta (HbBC). Any attempt to fit a Gaussian component will give us an error of the size of the rms. Even if we tried to put an upper limit, the flux would be x +/- 2x, which does not make sense. The fitting program would fix, in addition to the FWHM, the flux and probably the central lambda, to avoid taking the component to zero, with no improvement in the chi2.
-- Fig. 5, etc., and related text: As mentioned above, I stronlgy suggest to delete most mentions of variable NLR emission, since unphysical and very likely an artefact.
Done. We re-do Figure 5 and re-write the label.
-- line 203: It is very useful to estimate the SMBH mass of the system. It is rarely done for Sy types 1.8 and 1.9 because a large fraction of the BLR is believed to be obscured, and the continuum as well. So, a cautious comment on this might be given in a footnote.
It is still a useful order of magnitude estimate.
Done, we add a comment in the text and thank the referee for this suggestion.
It is fully up to the authors, but they could add a second SMBH mass estimate, based on sigma_[OIII] (as surrogate for stellar velocity dispersion), since they have these great optical spectra which cover [OIII] and even separately for both nuclei.
The problem is that using [OIII] as surrogate for sigma gives black hole mass estimations reliable within a factor of five. We prefer not to do these estimations.
-- Tab. 2: MEGARA, 2nd row entry: has error bars of +-0.0 assigned: a typo ?
Yes indeed, the typo was corrected.
-- Tab. 2: It would be interesting to inspect and briefly report the Balmer decrement; i.e., the flux ratio of broad Halfa/Hbeta. Deviations from the recombination value of ~3 could be used to measure extinction along the line of sight. Looks like no strong extinction, but some. But surprisingly and perhaps unexpectedly, the Balmer decrement is highest, when the Balmer lines are brightest....
Done. We added in Table 2 with our estimates of the Balmer decrement (BD) and a brief explanation of the obtained results in the 2nd paragraph of Sec.5. BD are shown in the last row of Table 2, and found that the BD also vary in time. This is an expected result in AGN with low Eddington ratios (e.g. see Fig. 2 of Wu+23).
-- Caption of Fig. 4, last sentence: To me, the red SDSS data point is still within (just at the border, within the errors) of the strong AGN regime. Again, because significant variability of a NLR is not expected on years timescale (but on 100s to 1000s if years), the NLR ionization should remain the same.
Agree. We have modified this on the label of Fig 4.
-- Sect. 7: Great idea to search for a second AGN in the second nucleus via the BLR emission. And I fully agree with the author's conclusion, that the identical spectra just imply seeing-based scattered emission from the brighter nucleus into the nearby regions.
Thanks.
-- line 301: see my earlier comment, that it is only the *optical continuum* that is observed (to be constant), while the lines are driven by the unobserved EUV continuum, etc.
So I suggest to leave open several possible scenarios: dynamical processes are certainly a possibility, but also alternatively accretion-disk changes affecting the unobservable EUV, or the possibility that the observed optical emission is host dominated (so the AGN continuum remains undetected even in the optical, and then could well be variable intrinsically).
Agree that there are several possible scenarios in the literature. A new paragraph includes more possible scenarios to explain the CL phenomenom in the discussion section.
Reviewer 2 Report
Comments and Suggestions for Authors
I judge this manuscript worthy of publication of a report of the observations after responding to the following minor comments:
1) There seems no description on what H-beta_BC and H-beta_NC designate.
2) There seems no explanation on the blue straight line in the top middle and top-right panels of Figure 4. What the black and red lines in the top panels of the same figure indicate had better be written in the figure caption.
Author Response
I judge this manuscript worthy of publication of a report of the observations after responding to the following minor comments:
1) There seems no description on what H-beta_BC and H-beta_NC designate.
Broad components (BC) and narrow components (NC) were described in the 2nd paragraph of the Introduction and in the fifth paragraph of section 4.
2) There seems no explanation on the blue straight line in the top middle and top-right panels of Figure 4. What the black and red lines in the top panels of the same figure indicate had better be written in the figure caption.
We described the Kewley and Kauffmann demarcation lines in the first paragraph of Section 5.1 and added a note in the Figure caption.
We used the redshift provided by the SDSS, which in turn uses all the emission lines in a wide range, from 3800 to 9000 AA to set the restframe. Furthermore, we added in the caption of Fig. 3 that we used the z from the SDSS.
Reviewer 3 Report
Comments and Suggestions for Authors
The authors present a detailed analysis of the spectra variability of the AGN Mrk 883, using several datasets showing the change of emission in the broad Balmer lines. After reviewing the paper, I recommend it to publication after some minor revisions. You can find below my points of concern:
1) It is not clear how the second nucleus identification was made when presented on the end of Sec. 1. Is it using the MEGARA FoV image on Fig. 1? What is the wavelength of this image? Could it be something else, maybe a clump of inflowing gas? Has this secondary nucleus been observed before, on x-rays perhaps?
2) On Sec. 3b, the authors mentioned that the blue and red MEGARA spectra were combined. Apparently the result of this combination is a decrease of spectral resolution from R~12000 to 5900. It seems also that the analysis performed in the work does not need for these two spectral regions to be combined, since they are essentially performed in separate. Could you please detail the reason for this combination, which in principle decreases the quality of your data?
3) Figs. 2 and 3: in general these figures should improve. Lines are quite thin, making it hard to differentiate, so please consider thicker lines. I also suggest adding a y-axis label. Residuals in fig. 3 are sometimes restricted to a small range in flux, so I suggest that you decrease the range accordingly to better visualize the behavior of the residuals. Also in fig. 3 I believe the authors switched the dashed and continuous line description on the caption, regarding residuals.
4) In general the emission lines seems to be redshifted in relation to the dashed line, representing the restframe wavelength. Is there a reason for this? In principle they should be centered at the same wavelength, since the redshift used to correct the spectra comes from SDSS as well, correct?
5) The scenario where outflows are suggested as an explanation for the broad lines variability is not very well explained in the last section of the paper. Is these outflows the same as observed on the [OIII] broader component? If yes, why is the variability so different (broad components peaking at MEGARA observations while [OIII] steadily increases)? Is there any other scenarios that could explain these variabilities?
Comments on the Quality of English Language1) I assume that the acronyms BC and NC mean “broad” and “narrow component”. However this is not defined anywhere in the text. Please do so.
2) Citations: The authors usually use simply [#], without explicitly mentioning the first author when citing. However sometimes they do use the first author, for example, in Benitez et al. [10]. I suggest you padronize the citations, with citations within the text mentioning the first author and in parentheses only with [#].
3) Typos:
-
Line 75: MrK 388 -> Mrk 388
-
Line 97: … OAN-SPM, hereafter these data will be … -> … OAN-SPM (hereafter these data will be …
-
Line 176: Hbeta(BC) -> Halpha(BC) ? (not sure if this is correct, but I assume the authors refer to a broad Halpha component here)
-
Line 200: blueshifted component for [OIII]. This …
Author Response
The authors present a detailed analysis of the spectra variability of the AGN Mrk 883, using several datasets showing the change of emission in the broad Balmer lines. After reviewing the paper, I recommend it to publication after some minor revisions. You can find below my points of concern:
1) It is not clear how the second nucleus identification was made when presented on the end of Sec. 1. Is it using the MEGARA FoV image on Fig. 1? What is the wavelength of this image? Could it be something else, maybe a clump of inflowing gas? Has this secondary nucleus been observed before, on x-rays perhaps?
The identification was done using the MEGARA spectral resolved data. From the spectral fitting analysis presented in section 7, we confirm that the emission of nuclei 2 has no AGN evidence in the optical spectrum. We will explore the nature of the second nuclei in a forthcoming paper focussed on our IFU data. The image of Figure 1 is an RGB composed image using the SDSS g-r-i bands. From the previous X-ray observations of Mrk 883, there is no mention of the second nuclei.
2) On Sec. 3b, the authors mentioned that the blue and red MEGARA spectra were combined. Apparently the result of this combination is a decrease of spectral resolution from R~12000 to 5900. It seems also that the analysis performed in the work does not need for these two spectral regions to be combined, since they are essentially performed in separate. Could you please detail the reason for this combination, which in principle decreases the quality of your data?
In section 7 we perform a full spectral fitting using BADASS3D, that models the AGN emission and the host spectra at the same time. To apply this analysis and minimize any degeneracy on the method, it is necessary to use a large spectral range, in addition the spectral resolution needs to be homogeneous. We add a more detailed text on section 7 to clarify this.
3) Figs. 2 and 3: in general these figures should improve. Lines are quite thin, making it hard to differentiate, so please consider thicker lines. I also suggest adding a y-axis label. Residuals in fig. 3 are sometimes restricted to a small range in flux, so I suggest that you decrease the range accordingly to better visualize the behavior of the residuals. Also in fig. 3 I believe the authors switched the dashed and continuous line description on the caption, regarding residuals.
Done, all these figures were improved.
4) In general the emission lines seems to be redshifted in relation to the dashed line, representing the restframe wavelength. Is there a reason for this? In principle they should be centered at the same wavelength, since the redshift used to correct the spectra comes from SDSS as well, correct?
We used the redshift provided by the SDSS, which in turn uses all the emission lines in a wide range, from 3800 to 9000 AA to set the restframe. We added in the caption of Fig. 3 that we used the z from the SDSS.
5) The scenario where outflows are suggested as an explanation for the broad lines variability is not very well explained in the last section of the paper. Is these outflows the same as observed on the [OIII] broader component? If yes, why is the variability so different (broad components peaking at MEGARA observations while [OIII] steadily increases)? Is there any other scenarios that could explain these variabilities?
The wind component is observed in the [OIII] and therefore is far away from the BLR. This wind has nothing to do with what we suggest can be the cause of the CL variability. For this behavior, different scenarios have been presented in the literature. We have add some of them in the discussion section. One of them suggests a dynamical origin related to the accretion disk that can produce a polar ionized wind, but is not the only one.
Comments on the Quality of English Language
1) I assume that the acronyms BC and NC mean “broad” and “narrow component”. However this is not defined anywhere in the text. Please do so.
Broad components (BC) and narrow components (NC) were described in the 2nd paragraph of the Introduction and in the fifth paragraph of section 4.
2) Citations: The authors usually use simply [#], without explicitly mentioning the first author when citing. However sometimes they do use the first author, for example, in Benitez et al. [10]. I suggest you padronize the citations, with citations within the text mentioning the first author and in parentheses only with [#].
This happened due to the style file of the journal. When using \citet the Surname appears, when using \citep only [#]. Anyway, we change this to have all references in the text only with [#].
3) Typos:
Line 75: MrK 388 -> Mrk 388
We checked, and the name is ok.
Line 97: … OAN-SPM, hereafter these data will be … -> … OAN-SPM (hereafter these data will be …
Done.
Line 176: Hbeta(BC) -> Halpha(BC) ? (not sure if this is correct, but I assume the authors refer to a broad Halpha component here)
Yes, it refers to the Broad Component.
Line 200: blueshifted component for [OIII]. This …
Done.